# Measurement Report: Carbonyl Sulfide production during Dimethyl Sulfide oxidation in the atmospheric simulation chamber SAPHIR

Marc von Hobe[1], Domenico Taraborrelli[2], Sascha Alber[1], Birger Bohn[2], Hans-Peter Dorn[2], Hendrik Fuchs[2], Yun Li[1,2], Chenxi Qiu[1], Franz Rohrer[2], Roberto Sommariva[3,4], Fred Stroh[1], Zhaofeng Tan[2], Sergej Wedel[2] and Anna Novelli[2]

[1]Institute for Energy and Climate Research, IEK-7: Stratosphere, Forschungszentrum Jülich GmbH, 52425 Jülich, Germany
[2]Institute for Energy and Climate Research, IEK-8: Troposphere, Forschungszentrum Jülich GmbH, 52425 Jülich, Germany
[3]School of Chemistry, University of Leicester, Leicester, UK
[4]School of Geography, Earth and Environmental Sciences, University of Birmingham, Birmingham, UK

*Correspondence to*: Marc von Hobe (m.von.hobe@fz-juelich.de)

**Abstract.** Carbonyl sulfide (OCS), the most abundant sulfur gas in the Earth's atmosphere, is a greenhouse gas, a precursor to stratospheric sulfate aerosol, and a proxy for terrestrial $CO_2$ uptake. Estimates of important OCS sources and sinks still bear significant uncertainties and the global budget is not considered closed. One particularly uncertain source term, the OCS production during the atmospheric oxidation of dimethyl sulfide (DMS) emitted by the oceans, is addressed by a series of experiments in the atmospheric simulation chamber SAPHIR at conditions comparable to the remote marine atmosphere. DMS oxidation was initiated with OH and/or Cl radicals and DMS, OCS and several oxidation products and intermediates were measured, including hydroperoxymethyl thioformate (HPMTF) that was recently found to play a key role in DMS oxidation in the marine atmosphere. One important finding is that the onset of HPMTF and OCS formation occurred faster than expected from the current chemical mechanisms. In agreement with other recent studies, OCS yields between 9 and 12 % were observed in our experiments. Such yields are substantially higher than the 0.7 % yield measured in laboratory experiments in the 1990s that is generally used to estimate the indirect OCS source from DMS in global budget estimates. However, we do not expect the higher yields found in our experiments to directly translate to a substantially higher OCS source from DMS oxidation in the real atmosphere, where conditions are highly variable and, as pointed out in recent work, heterogeneous HPMTF loss is expected to effectively limit OCS production via this pathway. Together with other experimental studies, our results will be helpful to further elucidate the DMS oxidation chemical mechanism and in particular the paths leading to OCS formation.

## 1 Introduction

In the Earth's atmosphere, sulfur compounds and aerosols are closely tied: under predominantly oxidizing conditions, sulfur gases are ultimately converted to sulfuric acid ($H_2SO_4$) that readily undergoes condensation to small aerosol droplets. With the exception of highly variable volcanic sources, the largest natural sulfur source to the atmosphere by far is oceanic emissions of dimethyl sulfide (DMS) with emission rates of about 28 Tg S a$^{-1}$ Lana et al. (2011). The importance of DMS emissions in

the context of tropospheric aerosols and clouds and their interactions with the Earth's climate system have been extensively discussed (Charlson et al., 1987; Park et al., 2021; Quinn and Bates, 2011; Sanchez et al., 2018). Barnes et al. (1994) have first pointed out the role of DMS as a source of carbonyl sulphide (OCS), the most important non-volcanic source of sulfate aerosol in the stratosphere that plays an important role for climate (Kremser et al., 2016).

## 1.1 The atmospheric OCS budget and the search for a "missing source"

Besides contributing $34 - 66$ Gg S a$^{-1}$ to the stratospheric sulfate aerosol layer (Kremser et al., 2016, and references therein) and being itself a greenhouse gas (Brühl et al., 2012), OCS is considered a valuable tracer to quantify carbon cycle processes (Whelan et al., 2018, and references therein). Its average tropospheric mixing ratio has been relatively constant around 500 ppt for the last four decades suggesting that sources and sinks are approximately balanced. This was in good agreement with a

bottom up OCS budget published by Kettle et al. (2002), but a revision of the largest OCS sink, i.e. the uptake by vegetation and soils (Berry et al., 2013; Sandoval-Soto et al., 2005; Suntharalingam et al., 2008), has led to a gap of a few hundred Gg S a$^{-1}$ in budget estimates. The OCS budget and its uncertainties have been reviewed in detail by Kremser et al. (2016) and Whelan et al. (2018), who discuss the possible contributions of marine and anthropogenic emissions and an overestimation of the terrestrial sink to the "missing OCS source".

Evidence that at least part of the missing source appears to be located over the ocean, and specifically over the tropical Pacific, comes from satellite observations (Glatthor et al., 2015) and inverse modelling studies (Kuai et al., 2015; Ma et al., 2021). OCS cycling in seawater is well understood and the direct oceanic OCS flux is sufficiently constrained, and its uncertainty is too small to explain a significant fraction of the "missing source" (Lennartz et al., 2021; Lennartz et al., 2017). The indirect OCS source from $CS_2$ emitted from the ocean and partially converted to OCS in the atmosphere is currently also considered

unlikely to fill the gap, although uncertainties in marine $CS_2$ cycling and emissions are larger than for OCS (Lennartz et al., 2021). Another indirect marine OCS source is the production during the DMS oxidation in the atmosphere.

## 1.2 Atmospheric DMS oxidation: a candidate for the missing OCS source?

In OCS budget estimates, the indirect source of OCS from atmospheric DMS oxidation has been calculated by multiplying the integrated DMS emission inventory (Lana et al., 2011) by a yield of 0.7 % for OCS produced from DMS determined in

laboratory experiments (Albu et al., 2008; Barnes et al., 1994, 1996). A higher OCS yield in the remote marine atmosphere has been speculated upon by Lennartz et al. (2017), and recent theoretical (Khan et al., 2021; Wu et al., 2015; Jernigan et al., 2022), laboratory (Jernigan et al., 2022; Ye et al., 2021; Ye et al., 2022; Berndt et al., 2019) and field studies (Veres et al., 2020; Vermeuel et al., 2020) provided new insights in the DMS oxidation mechanism, raising doubts about a globally constant yield of OCS production from DMS.

Based on theoretical calculations, the study by Wu et al. (2015) was the first to point out the potential importance of isomerization reactions in the DMS oxidation scheme under remote conditions at low concentrations of nitrogen oxides (NOx) and peroxy radicals (HO$_2$ and RO$_2$). Wu et al. (2015) proposed an isomerization rate constant of 2.1 s$^{-1}$ (293 K) for the primary

RO$_2$ formed following the OH abstraction of the methyl hydrogen (CH$_3$SCH$_2$O$_2$). This reaction would be the dominant loss process of the RO$_2$ leading to the formation of hydroperoxymethyl thioformate (HPMTF). Several laboratory studies have been carried out to measure this isomerization rate constant, finding values of 0.23 ± 0.12 s$^{-1}$ (295 K, Berndt et al., 2019), ~0.1 s$^{-1}$ (298 K, Jernigan et al., 2022) and 0.06 s$^{-1}$ (298 K, Assaf et al., 2023). Even with the slower isomerization rate of 0.06 s$^{-1}$, this reaction would still be the dominant reaction path in marine environments where nitric oxide (NO) mixing ratios are typically lower than 0.1 ppb. During the NASA Atom aircraft mission, Veres et al. (2020) measured HPMTF mixing ratios of up to 100 ppt and Vermeuel et al. (2020) observed up to 40 ppt in ground based measurements in the marine boundary layer, indicating the relevance of the suggested RO$_2$ isomerization pathway in the DMS oxidation in the real atmosphere.

The possibility of OCS formation following the reaction of HPMTF with OH was already suggested by Wu et al. (2015). Recent laboratory experiments (Jernigan et al., 2022; Ye et al., 2022) on the DMS oxidation by OH under low NOx/HO$_2$/RO$_2$ conditions clearly demonstrated the formation of the HPMTF intermediate in significant quantities and of OCS as a product with a yield of several percent, much higher than the yield of 0.7 % found in all previous experiments (Albu et al., 2008; Barnes et al., 1994, 1996), which were performed at much higher HO$_2$ and RO$_2$ concentrations, possibly preventing the isomerization reactions and the formation of HPMTF.

### 1.3 Aims of this study

Here, we present results from experiments investigating the DMS oxidation by OH and Cl radicals in the atmospheric simulation chamber SAPHIR. Experiments were performed with both artificial UV radiation sources as well as natural solar radiation at low concentrations of reagents to keep conditions as similar as possible to the marine environment (note that in the real atmosphere, marine aerosols are expected to affect the lifetimes of intermeidates including HPMTF; see, for example, Jernigan et al., 2022; Ye et al., 2022). Our approach of monitoring the decay of the DMS concentration and product formation after an initial DMS injection under varying conditions is complementary to the experiments of Jernigan et al. (2022), who employed a smaller dark chamber under a continuous flow regime. The experiments and instrumentation used to measure relevant species are described in Section 2. All measurement results are given in Section 3, which also discusses instrument intercomparisons for certain species and implications for data quality and the interpretation of the results. Constraints and open questions with respect to the chemical mechanism that are implied by our results are summarized and briefly discussed in Section 4.

## 2 Experimental

### 2.1 The atmospheric simulation chamber SAPHIR

Experiments were carried out in June 2020 in the atmospheric simulation chamber SAPHIR (Forschungszentrum Jülich), which has a cylindrical shape with a volume of 270 m$^3$. The SAPHIR chamber is designed for the investigation of oxidation processes under atmospheric conditions in a controlled environment. Its double walls, made of Teflon (FEP) film, are transmissive for the entire solar UV and visible spectrum (Bohn and Zilken, 2005), and a shutter system allows for sunlight exposure

or cut-off. Permanent flushing of the space between the double walls prevents diffusion of impurities from ambient air into the inner chamber, which is operated with synthetic air produced from evaporated liquid nitrogen and oxygen of highest purity (Linde, purity >99.99990 %). Two fans in the chamber ensure a complete mixing of trace gases within two minutes. The replenishment flow is controlled to maintain an overpressure of 35 Pa to avoid ambient air to penetrate the chamber, leading to a typical dilution of trace gases at a first order rate constant of approximately $1.5 \times 10^{-5}\,\mathrm{s^{-1}}$ (the actual dilution rate was continuously monitored both via the measured flow through the SAPHIR chamber and by the observed decay of $CO_2$ added to the chamber, see Supplementary Figure S7). More detailed descriptions of the SAPHIR chamber are given in earlier publications (Bohn et al., 2005; Rohrer et al., 2005).

**Table 1** Setup and conditions for each of the four experiments.

| | I | II | III | IV |
|---|---|---|---|---|
| | **8 June 2020** | **10 June 2020** | **12 June 2020** | **15 June 2020** |
| | Oxidation with OH @ high turnover rate | Oxidation with OH @ low turnover rate | Oxidation with $Cl_2$/Cl (in addition to OH) | Oxidation with OH @ high turnover rate |
| **Oxidant precursor injections** | two $O_3$ injections to ~60 and ~80 ppb | two $O_3$ injections to ~20 and ~18 ppb | $Cl_2$ flow for ~3:20 h one $O_3$ injection to ~120 ppb | one $O_3$ injection to ~55 ppb |
| **DMS injections** | one injection to ~16 ppb | two injections to ~16 and ~14 ppb | two injections to ~32 and ~38 ppb | one injection to ~28 ppb |
| **Lighting** | two UV-C lamps | one UV-C lamp | sunlight (roof open) | two UV-C lamps & sunlight |
| **$H_2O$ mixing ratio after humidification** | 8000 ppm | 16000 ppm | 7000 ppm | 18000 ppm |

## 2.2 Details of performed experiments

Four experiments were carried out to study the DMS oxidation and related OCS formation (Table 1). In all the experiments, the chamber was cleaned before the injection of trace gases by exchanging the chamber air 8 to 10 times with pure synthetic air. Evaporated Milli-Q® water was then introduced into the dark chamber by a carrier flow of synthetic air reaching mixing ratios given in Table 1. In Experiments II – IV, this was followed by injection of carbon dioxide ($CO_2$, Air Liquide N45, 99.995% purity, mixing ratio after the injection ~ 100 ppm) to improve spectral line locking of the OCS instrument.

To increase the OH radical concentration in the chamber in Experiments I, II and IV, OH was generated by the photolysis of added ozone (injected amounts are given in Table 1) and subsequent reaction of $O(^1D)$ that reacted with water vapour ($H_2O$). Radiation was provided by UV-C lamps (253.652 nm, Philips TUV 36W SLV/6) and ozone by a silent discharge ozonizer (O3onia). While the high $O_3$ absorption cross section of $1.1 \times 10^{-17}\,\mathrm{cm^2}$ at 254 nm allows for substantial photolysis and OH

production, other trace gases present in the experiments have absorption cross sections much smaller at 254 nm (DMS ~ 2 x $10^{-20}$ cm$^2$; OCS < 1 x $10^{-20}$ cm$^2$; SO$_2$ ~ 1 x $10^{-19}$ cm$^2$, no photolysis; DMSO < 1 x $10^{-21}$; HPMTF has no structure where strong absorption is expected as peroxides and aldehydes absorb little in that spectral range, i.e. on the order of $10^{-20}$ cm$^2$), and we do not expect the low intensity radiation of the lamps to photolyze any of these in a relevant amount. To prevent significant formation of nitrogen oxides (NO$_x$) from HONO, the lamps were the only light source, and the roof was closed for the entire

duration of Experiments I and II. The lamps were located in the central part of the chamber, which resulted in an inhomogeneous distribution of the radiation and OH radicals and therefore also of any trace gas produced from the reaction of OH with a lifetime shorter than the mixing time (~ 2 minutes). The trace gases relevant for this study (DMS, HPMTF, OCS and SO$_2$) have a much longer lifetime, so that a homogenous distribution can be assumed. Therefore, the determination of yields and measured concentrations of products were unaffected by the inhomogeneous illumination. Before turning on the lamps, DMS

(Sigma-Aldrich, natural, ≥ 99%) was introduced in the chamber reaching mixing ratios given in Table 1. In Experiment I, the lamps were on for ~ 8 hours while in Experiment II they were turned off after ~ 24 hours.

Experiment III was designed to study the DMS oxidation by chlorine radicals that may be relevant in the remote marine atmosphere under low OH conditions. After the injection of water vapour, DMS was injected (up to 40 ppb) and the chamber roof was opened. This allows for OH radicals to be produced from the photolysis of HONO, which is released by the chamber

wall upon illumination (Rohrer et al., 2005). After an hour of open roof, gaseous chlorine (Cl$_2$, Air Liquide, 10 ppm Cl$_2$ purity 99.8% and 1% CO$_2$ purity 99.995 in N$_2$ purity 99.999%) was continuously injected for ~ 3.5 hours at a rate of ~1.1 ppb h$^{-1}$. Its rapid photolysis produced a steady-state concentration of Cl radicals, competing with the OH for the reaction with DMS.

Experiment IV was similar to Experiments I and II: UV-C lamps were used to boost the production of OH radicals. In this experiment, DMS (up to ~ 30 ppb) was injected when the UV-C lamps were already turned on, but before the injection of O$_3$

to investigate, if DMS photolysis is relevant. After an hour, 50 ppb of O$_3$ were introduced. The lamps were turned on for ~ 8 hours. Three hours and 30 minutes after the lamp were switched on, the chamber roof was opened for 3 hours.

All experiments were designed such that chamber-specific sinks (dilution and wall loss of trace gases), and sources of trace gases that are formed in the sunlit chamber except for nitrous acid, did not influence the results.

## 2.3 Instruments

An overview of the analytical instruments and the parameters measured is given in Table 2. NO and NO$_2$ were measured by chemiluminescence (Eco Physics, TR780), CO, CO$_2$, CH$_4$, and H$_2$O by cavity ring-down spectroscopy (CRDS, Picarro, G2401), and O$_3$ by UV absorption (Ansyco-41M and Thermo scientific-49I). Photolysis frequencies inside the chamber were derived from solar actinic flux densities measured by a spectroradiometer mounted on the roof of the nearby institute building (Bohn et al., 2005; Bohn and Zilken, 2005). Formaldehyde (HCHO) was detected by CRDS (Picarro, G2307, Glowania et al.,

2021). DMS was monitored by a proton-transfer-reaction time-of-flight mass spectrometer (PTR-TOF-MS, Ionicon, Jordan et al., 2009), which unfortunately was not calibrated in our experiment for any other sulfur gases that were observed by PTR-MS in the DMS oxidation experiments by Jernigan et al. (2022) and Ye et al. (2022). Raw signals at m/z ratios where DMSO,

**Table 2**. Specification of instruments used in this study. Abbreviations are all spelled out in the text.

| Species | Measurement technique | Time resolution | Limit of detection ($1\sigma$) | $1\sigma$ accuracy |
|---|---|---|---|---|
| OH | LIF | 270 s | $0.3 \times 10^6$ cm$^{-3}$ | 13 % |
| OH | DOAS | 134 s | $0.8 \times 10^6$ cm$^{-3}$ | 6.5 % |
| $HO_2$, $RO_2$ | LIF | 47 s | $1.5 \times 10^7$ cm$^{-3}$ | 16 % |
| OH reactivity ($k_{OH}$) | LP-LIF | 180 s | 0.2 s$^{-1}$ | 10 % |
| Photolysis frequencies | Spectroradiometer | 60 s | | 10 % |
| $O_3$ | UV photometry | 60 s | 0.5 ppb | 2 % |
| $NO_X$ | Chemiluminescence | 60 s | NO: 20 ppt | NO: 5 % |
| ($NO+NO_2$) | | | $NO_2$: 30 ppt | $NO_2$: 7 % |
| CO, $CO_2$, $CH_4$, $H_2O$ | CRDS | 60 s | CO and $CH_4$: 1 ppb | 5 % |
| | | | $CO_2$: 25 ppb | |
| | | | $H_2O$: 0.1 % | |
| HCHO | CRDS | 300 s | 0.1 ppb | 10 % |
| HCHO | DOAS | 130 s | 0.3 ppb | 7 % |
| $Cl_2$ | I-CIMS | 60 s | 1 ppt | 5 % |
| DMS | PTR-TOF-MS | 30 s | 15 ppt | 14 % |
| OCS | OA-ICOS | 120 s | 15 ppt | 4 %[*] |
| $SO_2$ | DOAS | 60 s | 1 ppt | 5 % |
| $SO_2$ | CIMS | 300 s | 250 ppt | 40 %[#] |
| HPMTF/TPA | CIMS | 300 s | *not calibrated* | |

[*]30 ppt at OCS < 750 ppt   [#]500 ppt at $SO_2$ < 1250 ppt

DMSO$_2$, CH$_3$SCHO and CH$_3$SCH$_2$OOH would be expected to show up are shown in Supplementary Figures S1 – S4, but as these signals are not quantitative and may partly or even entirely be caused by interferences from different organic compounds, they are not considered in the results and discussion below. Cl$_2$ was detected by a chemical ionization mass spectrometer using iodine as reagent ion (I-CIMS, Sommariva et al., 2018; Tan et al., 2022). The concentrations of OH, HO$_2$ and RO$_2$ radicals were measured by a laser induced fluorescence (LIF) instrument permanently in use at the SAPHIR chamber and described

previously (Holland et al., 2003; Fuchs et al., 2011; Cho et al., 2022), but the LIF OH measurement is only used for Experiment III where the UV lamps were not used. The OH reactivity ($k_{OH}$), the inverse lifetime of OH, was measured by a pump and probe technique coupled with the time-resolved detection of OH by LIF (Lou et al., 2010; Fuchs et al., 2017). The OH radical together with SO$_2$ and HCHO were also measured by differential optical absorption spectroscopy (DOAS, Dorn et al., 1995; Glowania et al., 2021) in Experiment IV.

OCS was measured by off-axis-integrated cavity output spectroscopy (OA-ICOS, Baer et al., 2002; O'keefe, 1998; Paul et al., 2001) using a prototype of a commercially available ABB Los Gatos OCS analyser (more details on the instrument are given in Kremser et al., 2021). Air was directly sampled from the SAPHIR chamber through ~8 m 3/8" OD Teflon (PTFE) tubing at a mass flow rate of $6 \times 10^{-6}$ kg s$^{-1}$. To ensure data quality and determine measurement accuracy and precision particularly for OCS, calibrations were carried out prior to the experiments using calibration standards providing mixing ratios of 0.25–5 ppb

prepared with a permeation system as well as a certified standard containing 450 ppt OCS (National Oceanic and Atmospheric Administration NOAA, Boulder, USA).

The FZ-Jülich IEK-7 project FunMass stands for a suite of Chemical Ionisation Mass Spectrometers (CIMS) employing a custom-made ion-funnel in the ion-molecule-reaction (IMR) region (operating at around 35 hPa) and employing TOFWERK Time-of-Flight mass analysers as described in Albrecht (2014) and Khattatov (2019). Here we used the laboratory version

FunMass-L employing iodide ion chemistry and a high-resolution H-ToF mass analyser (e.g. Lee et al., 2014). Employing a 0.4 mm diameter PTFE nozzle it was sampling a 1.1 SLM (STP liter per minute) flow from the SAPHIR chamber through a 6 mm outer diameter PFA tubing with a length of 62 cm from the inlet into the IMR which was operating at 35 hPa. Reagent iodide ions were produced by passing methyl iodide (CH$_3$I) mixed in N$_2$ (Linde, purity >99.9999 %, ca. 1 ppm CH$_3$I, 0.25 SLM) through a radioactive ion source (Po-210, NRD P-2021, 370MBq). The iodide ions and their water clusters then react

with relevant analyte species like HPMTF and TPA (thioperformic acid) to form iodide ion-molecule clusters detected in the mass spectrometer at mass-to-charge ratios (m/Q) of 234.893 and 204.883, respectively (Berndt et al., 2019; Veres et al., 2020; Ye et al., 2022). Typical residence time in the IMR is 60 ms.

Any major interference of the (I*HPMTF)$^-$ cluster by the isobaric (I*N$_2$O$_5$)$^-$ cluster is ruled out for the following reasons. A zero-dimensional box model constrained to observed NO, NO$_2$ (up to 1.5 ppb), O$_3$ and temperature was run to simulate ex-

pected concentrations of NO$_3$, HNO$_3$ and N$_2$O$_5$ in the chamber. This resulted in N$_2$O$_5$ mixing ratios well below 1 ppt, which is rather low such that a significant interference on the HPMTF signal can be excluded. Further, the similar temporal evolution of the reported HPMTF and TPA signals, which differ substantially from the modelled temporal profile of N$_2$O$_5$, also support

that there is no major interference. We also note that modelled $HNO_3$ exhibits a very similar temporal evolution as the uncalibrated I-CIMS signal for the $(I*HNO_3)^-$ cluster giving further confidence in the modelled $N_2O_5$ evolution. We also tried to explore the isotopologue signatures for the HPMTF and the $N_2O_5$ clusters. The exercise was done for measurement day II. The 237 and 235 m/z peaks (the HPMTF $^{34}S$ and $^{32}S$ clusters, respectively) show a very similar temporal evolution, however, the average ratio found for the areas was very noisy and about 4-fold the expected 4.2 %, indicating that at least the 237 m/z peak is blended by some isobaric unknown ion. Therefore, this method could not be used to ensure the absence or low significance of $N_2O_5$. For the 236 m/z peak (the blended $^{33}S$-HPMTF (0.8 %) and $^{15}NNO_5$ (0.4 %) cluster) no significant peak was detected at all, which due to the generally low signal to noise ratios (S/N) also cannot rule out some contribution by $N_2O_5$. This applies to days III and IV as well since S/N is very similar.

For the $SO_2$ quantification we choose the $SO_5^-$ ion cluster observed at m/Q=111.9968 (e.g. Möhler et al., 1992; Seeley et al., 1997) formed from the reaction of $SO_2$ and $CO_3^-$. The latter ion was used for normalization and was most probably formed from varying amounts of $CO_2$ in the IMR. The formation of $SO_5^-$ by reactions of $IO_x^-$ (x=2, 3, 4) or $O_2^-$ as described by Eger et al. (2019) seems very improbable due to extremely low abundances of these potential reagent ions. The possible origin of the $CO_3^-$ ions from $O_2^-$ chemistry in a two-step process involving $CO_2$ and $O_3$ as described by Novak et al. (2020, see R4a/b and R5a/b) also seems rather improbable due to the very low $O_2^-$ signals involved. This is supported by the absence of any correlation of the $CO_3^-$ and $SO_5^-$ signals even for significant changes in $O_3$ as, e.g., encountered for the second ozone addition (see Fig. 2 around 8.5 h). Unfortunately, a detailed pathway for the $CO_3^-$ formation cannot be given. The iodide cluster $I*SO_2^-$ could not be detected with our setup, most probably due to the electric fields generated within the ion funnel leading to collision induced dissociation.

Online calibration was performed for $SO_2$ employing a compact permeation oven containing a gravimetrically quantified $SO_2$ permeation tube (Fine Metrology, Italy) as described by Von Hobe et al. (2023). Regular additions from the oven adding 700 ppt $SO_2$ to the IMR analyte gas (either chamber air or synthetic air) were done. However, due to the varying and weak $CO_3^-$ reagent ion abundance measured $SO_2$ mixing ratios mostly cannot be statistically quantified to better than around 40 %. We referenced all $SO_2$ data to the starting signal (ncts) once humidification and $CO_2$ injection were completed and signals stabilized (days III and IV). For day II due to a late measurement start of the CIMS (just before turning on UV lamps and about one hour after the DMS addition) the reference signal was derived from the very first measurement points somewhat later than for days III and IV. However, the $SO_2$ reference signals which corresponded to about 200 ppt agreed within about +/- 100 ppt for all days. This is well below the other main error sources. We cannot assign the origin of this background. Presented $SO_2$ therefore represented the increase from the start of the DMS oxidation which corresponds to the reacted DMS. Uncertainties resulting from this procedure mainly due to changing humidity and ozone are estimated to a maximum of +50 and -350 ppt of $SO_2$, respectively. However, the comparison to the DOAS instrument shows a quite good agreement of better than 10 % for experiment IV when both measurements were operating (Fig. 4) giving further confidence into the applied procedures.

Calibrations for HPMTF or TPA were not carried out for this campaign, therefore just the ratios of counts for detected product ions over the sum of the $I^-$ and $I*H_2O^-$ signals are reported here. The change in this quantity, here termed normalized counts

or ncts, is to first order proportional to concentrations of the neutral analytes (e.g. Huey, 2007). However, missing a proper way to scrub the relevant species detected by the CIMS while retaining ozone and humidity, we could not establish a proper zero background here. For the HPMTF and TPA measurements this procedure did not produce consistent starting values and we report none-background corrected values. These background values before expected presence of OH (UV light for days II and III and $O_3$ for day IV) where below 20% of the maximum observed signals also for day II with the late measurement start of the CIMS. Clearly, we cannot exclude preexisting HPMTF or TPA at these starting points. Therefore, in our view it is also not possible to establish a HPMTF formation from DMS and ozone alone based on the available data. Also, some background variations expected due to variations in relative humidity, which usually dropped slowly but significantly over the experiments (up to 35% over a 8h measurement, see Supplementary Fig. S2) cannot be ruled out.

## 3 Results and discussion

Time series of observed radicals, trace gases, stable intermediates and inferred reaction rates that are discussed in some detail in this section are shown for the respective experiments in Figures 1 – 4, with additional parameters and concentrations of further gas-phase compounds and radicals given in the Supplementary Figures S1 – S4 (the time zero used for the relative time scale in all figures is defined for each experiment by the respective first DMS injections). Below, we describe the important observations for each experiment.

### 3.1 Experiment I: high turnover rate of DMS

In Experiment I, $O_3$ was added to reach mixing ratios between 50 and 80 ppb and both UV-C lamps were turned on to produce OH (Figure 1a). With $HO_2$ and $RO_2$ concentrations of ~ 1.5 x $10^9$ and 3 x $10^9$ cm$^{-3}$, respectively, and no detectable NO present (Supplementary Figure S1), the isomerization reaction would be the dominant loss process for $CH_3SCH_2O_2$ similar to what is expected in pristine marine air. The DMS loss rate constant, determined by taking the derivative of the observed DMS decay, slowly rises from ~7 x $10^{-5}$ s$^{-1}$ to ~1.4 x $10^{-4}$ s$^{-1}$ (Figure 1c). Using the combined DMS + OH rate constants at 298 K for the H-abstraction and OH addition pathways of 6.4 x $10^{-12}$ cm$^3$ molecules$^{-1}$ s$^{-1}$ (Barnes et al., 2006, and references therein) and assuming that DMS is consumed only by the OH radical and dilution, this would be consistent with an average OH concentration in the range of 1 – 2 x $10^7$ molecules cm$^{-3}$. Nearly all the DMS injected had reacted after about 9 hours (Figure 1b). Effects from the observed temperature variations of ~10 °C in Experiment I (see Supplementary Figure S1c) on the DMS + OH reactions are impossible to discern due to the low precision of the observed DMS decay and lacking observations of the OH concentration and variability. Nevertheless, it should be noted that while the rate constant of the DMS + OH abstraction reaction has a moderate temperature dependence (< 5 % over temperature range of 16 – 35 °C over all four experiments, see Supplementary Figures S1 – S4), the rate constant of the DMS + OH addition reaction changes from 3.3 x $10^{-12}$ at 16 °C to 1.3 x $10^{-12}$ at 35 °C (Burkholder et al., 2019). This means that at the lower end of the temperature range in our experiments, we expect about 60 % to react through abstraction and 40 % through addition, while at the upper end, it is more like 80 % going

through abstraction and 20 % through addition. As a consequence, we would expect more OCS production at higher temperatures as we get more HPMTF from getting more of the primary $RO_2$ and a faster isomerization rate due to the temperature.

On the product side, CO starts to increase as soon as DMS and OH are available (Figure 1b). Towards the end of the experiment, observed CO levels exceed the original amount of DMS. While a CO yield of up to 2 is theoretically possible because of the two carbon atoms in DMS, additional CO production from the oxidation of hydrocarbons outgassing from the chamber film cannot be ruled out. As shown in Figure 1c) HCHO increases rapidly during the first three hours after DMS and OH are available, and then starts to level off and later decreases, showing that at this point, the removal exceeds the production. OCS

concentrations increase almost linearly at an average rate of $\sim 2 \times 10^6$ molecules $cm^{-3}$ $s^{-1}$ until DMS is nearly consumed (Figure 1 b and c). An interesting observation is the lack of any substantial time delay (i.e. more than ~15 minutes) between the start of DMS oxidation and OCS production, indicating that this must at least partly proceed via reactions involving only short-lived intermediates.

The total OCS yield per DMS consumed in Experiment I was $10.4 \pm 1.6$ % (the given uncertainty is propagated from the

260 uncertainties in chamber dilution rate and the OCS and DMS measurements). This was calculated by dividing the amount of OCS present at the end of the experiment by the dilution corrected total sulfur in the chamber (any sulfur was present as DMS at the beginning of the Experiment, see Supplementary Figure S5) minus the DMS still present:

$$\Phi_{OCS}(t) = \frac{[OCS](t)}{\sum_{DMS\ additions}[DMS]_{init} \times (1 - k_{SAPHIR})^{t-t_{init}} - [DMS](t)} \tag{1}$$

where $k_{SAPHIR}$ is the observed chamber dilution rate and the summation is carried out for each DMS injection. This, as well the

265 OCS yields given in the following Sections, are experimental results that strongly depend on the conditions in the SAPHIR chamber. We expect OCS yields from DMS oxidation in the real atmosphere to be significantly lower, as will be discussed in Section 4.

### 3.2 Experiment II: low turnover rate of DMS

As a result of $O_3$ mixing ratios being held between 8 and 20 ppb (significantly less than in Experiment I) and only one UV-C

lamp turned on to photolyze it, OH concentrations in Experiment II are expected to be significantly lower than in Experiment I, which is supported by the lower DMS removal that never exceeded $\sim 6 \times 10^{-5}$ $s^{-1}$ in this experiment (Figure 2d). After about 14 hours, more than 50 % of the sulfur present in the chamber is still present as DMS (see Supplementary Figure S6b).

Products are also formed at lower rates than in Experiment I, but the qualitative behaviour is similar. In addition to CO, HCHO and OCS, measurements of $SO_2$, as an expected major sulfur product of the DMS oxidation, are also available in this experi-

275 ment (Figure 2b). By the end of the experiment, about 2.5 ppb $SO_2$ are observed making up most of the converted sulfur still in the chamber (for a sulfur budget of all measured sulfur compounds, see Supplementary Figure S6). The yield of OCS produced in Experiment II was $11.5 \pm 1.8$ % (calculated according to Equation 1).

HPMTF and TPA, proposed as important intermediates for OCS production from DMS by Jernigan et al. (2022), both increase rapidly after the onset of DMS removal (Figure 2c). The HPMTF concentration peaks after about an hour and again an hour

after the seconds DMS injection, suggesting its removal to proceed fairly rapidly. As explained in Section 2.3, the measurements of these intermediates are not calibrated and only relative signals are available.

### 3.3 Experiment III: Oxidation of DMS by chlorine

In Experiment III DMS was additionally oxidized by Cl (expected to be present in the marine environment) instead of only OH like in Experiments I and II. However, as stated in Section 2.1, OH is always produced from the photolysis of HONO coming off the chamber film as soon as the chamber roof is opened (Figure 3a). At an OH concentration of $\sim$1 x $10^7$ molecules cm$^{-3}$, the DMS removal rate from reaction with OH is expected to be 6.4 x $10^{-5}$ s$^{-1}$. The observed rate (Figure 2d) is about twice as high during the Cl$_2$ addition phase (3:20 hours, starting 1 hour after the first DMS addition), indicating that reaction with either Cl$_2$ (Dyke et al., 2005; Dyke et al., 2006) or Cl radicals contributes significantly to the removal of DMS.

The observed OCS production rate of up to 7 x $10^6$ molecules cm$^{-3}$ s$^{-1}$ during this period (Figure 3d) is higher than in any other experiment, so that it seems likely that OCS is being produced not only when DMS oxidation is initiated by OH but also when Cl$_2$ and/or Cl are involved as initial oxidizing agents. Differently from the other experiments in this work, the concentration of NO was not negligible in Experiment III (Supplementary Figure S3b) reaching up to $\sim$ 200 ppt before the ozone injection and decreasing to $\sim$ 50 ppt afterwards. Even during the part of the experiment with high NO mixing ratios, large OCS levels are observed consistent with a unimolecular isomerization rate constant for CH$_3$SCH$_2$O$_2$ faster than $\sim$ 0.05 s$^{-1}$ (298 K).

HPMTF and TPA are both present at measurable levels (Figure 3c), but because a quantitative comparison between experiments is not possible here, it cannot be resolved if they are exclusively produced via the OH oxidation chain or also via reactions involving chlorine. Mechanistically, HPMTF production following H-abstraction from DMS upon reaction with Cl can be expected to proceed like after H-abstraction with OH. Nothing is known about the secondary chemistry when HPMTF reacts with Cl and how this may promote or inhibit OCS producing channels. With the open roof in Experiment III, photolysis reactions may also play a role in HPMTF chemistry (see Khan et al., 2021, for a discussion of HPMTF photolysis reactions). An interesting observation is the drop in OCS production (Figure 3d) when O$_3$ is added later in the experiment. With the rise of HPMTF and TPA levels related to the second DMS injection just prior to O$_3$ addition, one might expect OCS production to continue at the same rate or faster. We can only speculate that ClO, which is rapidly formed when both Cl and O$_3$ are present, competes as reaction partner in one or more steps in the DMS oxidation chain with impacts on the distribution of intermediates and products. It should be noted that the Cl + O$_3$ reaction is about ten times slower than the Cl + DMS reaction and a maximum ClO concentration of 4 x $10^7$ molecules cm$^{-3}$ is estimated from steady state calculations. Because ClO is less reactive towards DMS than Cl (ClO reacts with DMS at a rate constant of $9.5 \pm 2.0 \times 10^{-15}$ cm$^3$ molecule$^{-1}$ s$^{-1}$ according to Barnes et al., 1991), it is not expected to be competitive with Cl and even Cl$_2$ for the oxidation of DMS. The overall OCS yield at the end of Experiment III was $9.5 \pm 1.5$ % (calculated according to Equation 1).

## 3.4 Experiment IV: high turnover rate of DMS and sunlight

The initial conditions in Experiment IV were similar to those in Experiment I, only with higher DMS mixing ratios. Later in the experiment, the chamber roof was also opened, allowing for HONO photolysis as an additional OH source and, like in Experiment III, for photolysis reactions in the visible and near UV. Another difference to the Experiments I – III is the availability of measurements by the DOAS instrument. $SO_2$ mixing ratios measured by the DOAS were about 20 – 30 % lower than those observed by FunMass (Figure 4b). The DMS decay rate constant calculated from the OH radical concentrations measured by the DOAS instrument was in good agreement with the observed DMS concentration time series within the measurement uncertainties (shown in the Supplementary Figure S5) as can be expected because the DOAS instrument measures along the centreline of the chamber.

Product and intermediate formation behaved similar as in the Experiments I – III. Production of HPMTF, TPA and OCS commences quasi immediately after DMS was added, with both HPMTF and TPA concentrations peaking within about 30 minutes after the DMS addition (Figure 4c). Assuming that (i) HPMTF formation dominates over bi-molecular reactions so that HPMTF forms approximately at the rate of DMS removal via the H-abstraction reaction (strictly, this is an upper limit), and that (ii) HPTMF is only lost by its reaction with OH and dilution, expected HPMTF mixing ratios are calculated for different values of the reaction rate constant $k_{HPMTF+OH}$, shown in Figure 4 c). Adjusting $k_{HPMTF+OH}$ so that the shape of the calculated HPMTF time series fits the shape of the observed ion mass signal of HPTMF gives a value of the reaction rate constant of around $5 \times 10^{-11}$ $cm^3$ molecules$^{-1}$ s$^{-1}$ (Fig. 4c), higher than the values of $k_{HPMTF+OH} = 1.4 \times 10^{-11}$ $cm^3$ molecules$^{-1}$ s$^{-1}$ determined by (Jernigan et al., 2022) and the value of $k_{HPMTF+OH} = 2.1 \times 10^{-11}$ $cm^3$ molecules$^{-1}$ s$^{-1}$ (Ye et al., 2022). Because HPMTF was only measured in relative terms and not as absolute mixing ratio and because this steady-state approach depends on two critical assumptions that may not hold 100 %, one should be cautious and not interpret this result as a quantitative determination of $k_{HPMTF+OH}$. A drop in HPMTF below the first order removal curve is observed when the chamber roof is opened. As much of the DMS has already been removed by this time, this drop would be difficult to explain by a sudden change in the HPMTF production rate (e.g. from competing binary reactions) and is more likely caused by a faster/additional removal process such as daylight photolysis. This is, however, only speculative, and further experiments to corroborate and quantify this are warranted. The dilution corrected overall OCS yield in Experiment IV was $11.9 \pm 1.8$ % (calculated according to Equation 1).

## 4 Conclusions

In all four experiments, significant OCS production was observed with total yields between 9 and 12 %. This is significantly higher than the 0.7 % observed in earlier experiments (Albu et al., 2008; Barnes et al., 1994, 1996) and in the same range as recently observed by Jernigan et al. (2022), who used a rather different experimental design and chamber, which gives confidence to both results. It needs to be stressed that such experimental OCS yields from DMS oxidation will not necessarily pertain to the real atmosphere. First, low NOx conditions favouring the isomerization reaction of $CH_3SCH_2O_2$ resulting in

HPMTF formation over bi-molecular reactions are only found in pristine marine environments. Second, as pointed out by Jernigan et al. (2022), heterogeneous loss of HPTMF will suppress OCS formation in the real atmosphere. As a result, substantial variability in OCS formation during DMS oxidation is expected, and back-of-the-envelope type calculations using a

single OCS yield to derive a global indirect OCS source from global DMS emission estimates are not warranted. Rather, a robust estimate of this OCS source requires detailed parameterization in atmospheric models, and for that, a full qualitative and quantitative understanding of the OCS production mechanism and of other atmospherically relevant HPMTF loss processes is needed. Based on some of our observations, future experiments to fully establish and quantify this mechanism should probably include HPMTF photolysis and the role of chlorine radicals as a potentially relevant oxidant under certain atmos-

pheric conditions.

Observations by the FunMass instrument corroborate findings of Jernigan et al. (2022) that HPMTF appears to be a key intermediate in the DMS oxidation chain and most likely for the OCS forming channels. Lacking a robust quantification method for HPMTF, it is impossible to quantitatively derive its formation and decay rates from our experiments. The appearance of OCS within minutes after the start of the DMS oxidation process and the rapid increase and decay of the HPMTF signal may

indicate that this chemistry proceeds faster than expected from the current state-of-the-art mechanistic theory as described e.g. in Jernigan et al. (2022) as "multi generation mechanism". The relative increase in the HPMTF decay rate when the roof is opened in Experiment IV suggests that HPMTF photolysis reactions could play a significant role at least under the given experimental conditions. Clearly, these results are only qualitative and should be viewed with caution given the relative HPMTF scale and the uncertainties on the level and homogeneity of the OH concentration in the SAPHIR chamber in our

experiments. TPA, an intermediate thought to play a role further down the reaction chain, was also observed, again in qualitative agreement with the experiment by Jernigan et al. (2022). As for HPMTF, the shape of the TPA trace suggests rapid formation and removal reactions.

**Data Availability**

Data from the experiments in the SAPHIR chamber used in this work are available on the EUROCHAMP database web page

(https://data.eurochamp.org/). Data for each experiment are available as follows: experiment on 08 June 2020 (Experiment I), Novelli et al. (2023a, https://doi.org/10.25326/667N-KA95); experiment on 10 June 2020 (Experiment II), Novelli et al., 2023b, https://doi.org/10.25326/74PS-NM77); experiment on 12 June 2020 (Experiment III), Novelli et al. (2023c, https://doi.org/10.25326/57DX-WR36); and experiment on 15 June 2020 (Experiment IV), Novelli et al. (2023d, https://doi.org/10.25326/BYAV-YK31).

## Author Contributions

MvH, FS, YL, SA, CQ, FR, HF, HPD, SW, ZT, RS, BB and AN collected and quality checked the data. MvH and AN wrote the manuscript. AN and DT designed and lead the experiments. All authors were involved in helpful discussion and contributed to the manuscript.

## Competing interests

Marc von Hobe is member of the editorial board of Atmospheric Chemistry and Physics. The peer-review process is being guided by an independent editor, and the authors have also no other competing interests to declare.

## Acknowledgements

Sascha Alber, Yun Li, and Chenxi Qiu were supported by the graduate school HITEC of Forschungszentrum Jülich.

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

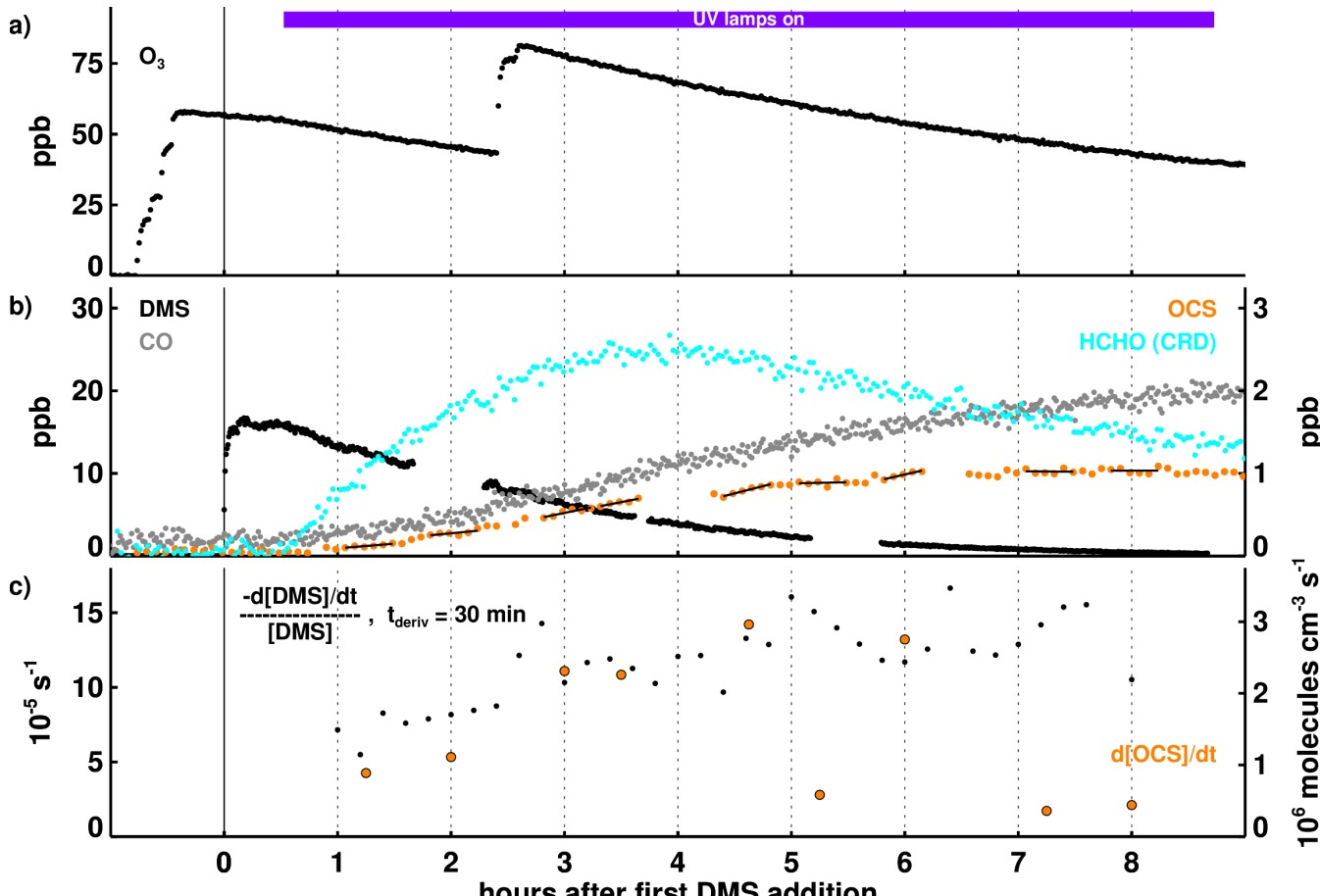

**Figure 1** Time series of selected parameters in Experiment I, with time in hours after the first DMS injection and the period with the UV
lamps turned on indicated by the coloured bar on the top of panel a). In panels a) and b), mixing ratios for trace gases are shown with the
position of the labels indicating which axis refers to which species. In panel c), the first order DMS decay rate is determined for each
individual point by a linear least squares fit of the DMS concentration data within ± 15 minutes of this point. OCS production rates given in
panel c) are calculated by linear least squares fitting of the OCS concentration data marked with the black lines in panel b), each encompass-
ing 30 minutes of data (the time periods were chosen to avoid data gaps and periods where the production rate changed significantly within
30 minutes). Both DMS loss and OCS formation rates shown are corrected for the loss induced by the replenishment flow in the SAPHIR.

585

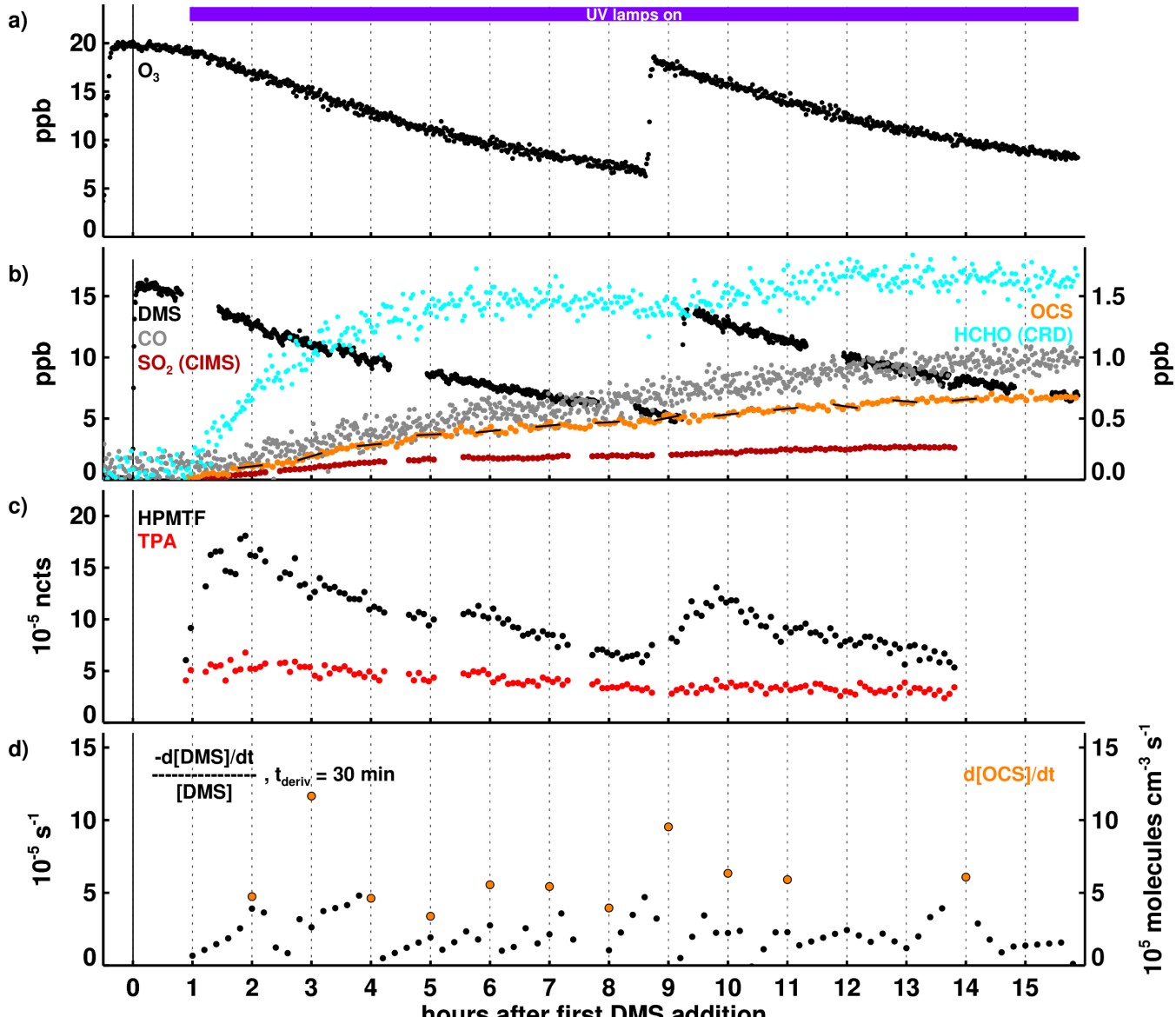

**Figure 2** Time series of selected parameters in Experiment II, with time in hours after the first DMS injection and the period with the UV lamps turned on indicated by the coloured bar on the top of panel a). In panels a) and b), mixing ratios for trace gases are shown with the position of the labels indicating which axis refers to which species. In panel c), HPMTF and TPA are shown as normalized counts (see Section 2.3). In panel d), the first order DMS decay rate is determined for each individual point by a linear least squares fit of the DMS concentration data within ± 15 minutes of this point. OCS production rates given in panel d) are calculated by linear least squares fitting of the OCS concentration data marked with the black lines in panel b), each encompassing 30 minutes of data (the time periods were chosen to avoid data gaps and periods where the production rate changed significantly within 30 minutes). Both DMS loss and OCS formation rates shown are corrected for the loss induced by the replenishment flow in the SAPHIR.

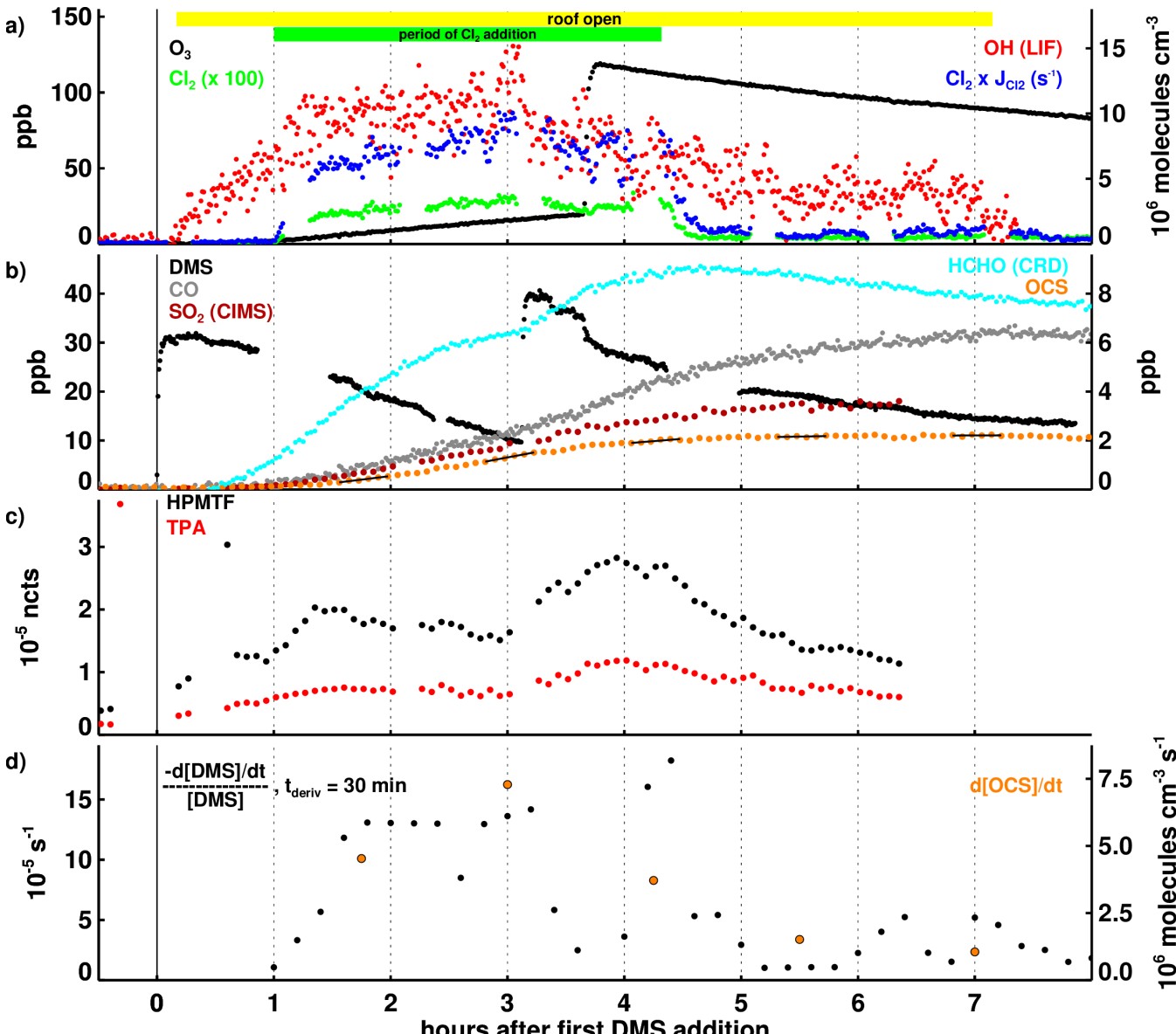

**Figure 3** Time series of selected parameters in Experiment III, with time in hours after the first DMS injection and the periods of daylight irradiation and continuous $Cl_2$ injection indicated by the coloured bars on the top of panel a). In panels a) and b), mixing ratios for trace gases, the OH concentration and the product of the $Cl_2$ concentration and its photolysis rate are shown with the position of the labels indicating which axis refers to which species. In panel c), HPMTF and TPA are shown as normalized counts (see Section 2.3). In panel d), the first order DMS decay rate is determined for each individual point by a linear least squares fit of the DMS concentration data within ± 15 minutes of this point. OCS production rates given in panel d) are calculated by linear least squares fitting of the OCS concentration data marked with the black lines in panel b), each encompassing 30 minutes of data (the time periods were chosen to avoid data gaps and periods where the production rate changed significantly within 30 minutes). Both DMS loss and OCS formation rates shown are corrected for the loss induced by the replenishment flow in the SAPHIR.

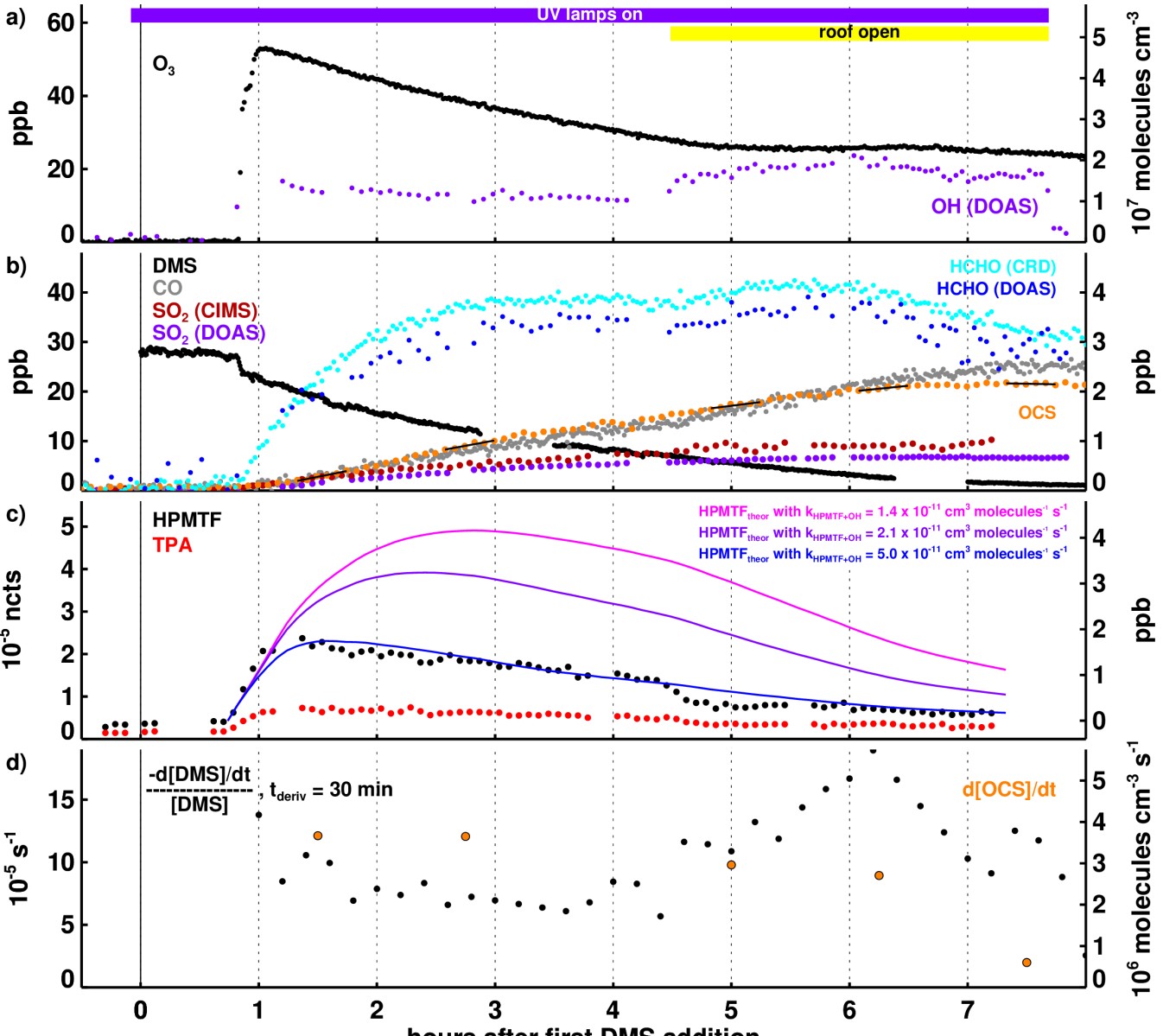

**Figure 4** Time series of selected parameters in Experiment IV, with time in hours after the first DMS injection and the periods with the UV
lamps turned on and the SAPHIR daylight shutters being open indicated by the coloured bars on the top of panel a). In panels a) and b),
mixing ratios for trace gases and the OH concentration are shown with the position of the labels indicating which axis refers to which species.
In panel c), observed HPMTF and TPA are shown as normalized counts (left side axis) and estimated (see text in Section 3.4) HPMTF
mixing ratios (right side axis) are shown for $k_{HPMTF+OH} = 1.4 \times 10^{-11}$ cm$^3$ molecules$^{-1}$ s$^{-1}$ (Jernigan et al., 2022, magenta), $k_{HPMTF+OH} = 2.1 \times$
$10^{-11}$ cm$^3$ molecules$^{-1}$ s$^{-1}$ (Ye et al., 2022, purple) and $k_{HPMTF+OH} = 5 \times 10^{-11}$ cm$^3$ molecules$^{-1}$ s$^{-1}$ (blue.) In panel d), the first order DMS decay
rate is determined for each individual point by a linear least squares fit of the DMS concentration data within ± 15 minutes of this point.
OCS production rates given in panel d) are calculated by linear least squares fitting of the OCS concentration data marked with the black
lines in panel b), each encompassing 30 minutes of data (the time periods were chosen to avoid data gaps and periods where the production
rate changed significantly within 30 minutes). Both DMS loss and OCS formation rates shown are corrected for the loss induced by the
replenishment flow in the SAPHIR.