# Peer review of "Measurement Report: Carbonyl Sulfide production during Dimethyl Sulfide oxidation in the atmospheric simulation chamber SAPHIR"

_EGUsphere, 2023_

## Author Comment (AC1)

We thank the reviewer for the positive feedback and the appreciation of our work.

Below, we address the general and specific comments and suggestions point by point. The review comments are repeated here in black, our replies are given in blue.

I would recommend the elaboration and restructuring of the work before final publication. Generally, I would stress the importance of oxidative and environmental drivers of OCS production from DMS oxidation throughout the work as well as the role of chloride and photolysis within DMS oxidation.

This is largely addressed by changes made in response to the specific comments below. We also add the following sentence in the conclusions:

> Based on some of our observations, future experiments to fully establish and quantify this mechanism should probably include HPMTF photolysis and the role of chlorine radicals as a potentially relevant oxidant under certain atmospheric conditions.

I would also recommend remaking the figures with updated legends to help the reader better understand which time traces are which.

We rework all figures to improve quality and readability.

Specific Comments

Line 112: A clarification of why photolysis is not a driving reaction would be greatly appreciated. 254 nm light is a higher energy light. Do you assume insignificant due to OH concentration or low light flux?

We add the following in Section 2.2 that should clarify this:

> While the high $O_3$ absorption cross section of $1.1 \times 10^{-17}$ cm$^2$ at 254 nm allows for substantial photolysis and OH production, other trace gases present in the experiments have much smaller absorption cross sections at 254 nm (DMS $\sim 2 \times 10^{-20}$ cm$^2$; OCS $< 1 \times 10^{-20}$ cm$^2$; SO$_2$ $\sim 1 \times 10^{-19}$ cm$^2$, not photolyzing; DMSO $< 1 \times 10^{-21}$; HPMTF has no structure where strong absorption is expected as peroxides and aldehydes absorb little in that spectral range, i.e. on the order of $10^{-20}$ cm$^2$), and we do not expect the low intensity radiation of the lamps to photolyze any of these in a relevant amount.

Table 2: TPA or Thioperformic acid is not referenced before this point. Would recommend adding the full name of the species before adding the abbreviation.

In the revised manuscript, the full name is included in the text describing the instrument.

Line 140: Ye et al and Jernigan et al found other DMS related products within their PTR-MS. Were any additional sulfur species found using the PTR implemented in this chamber experiment. Even uncalibrated signal could help understand when potential intermediates react to the perturbations within the chamber.

We add traces for uncalibrated signals in the PTR-MS data that potentially corresponded to relevant species (DMSO, $DMSO_2$, $CH_3SCHO$, $CH_3SCH_2OOH$) in the supplementary Figures (S1 – S4) and add the following statement in Section 2.3:

> DMS was monitored by a proton-transfer-reaction time-of-flight mass spectrometer (PTR-TOF-MS, Ionicon, Jordan et al., 2009), which unfortunately was not calibrated in our experiment for any other sulfur gases that were observed by PTR-MS in the DMS oxidation experiments by Jernigan et al. (2022) and Ye et al. (2022). Raw signals at m/z ratios where DMSO, $DMSO_2$, $CH_3SCHO$ and $CH_3SCH_2OOH$ would be expected to show up are shown in Supplementary Figures S1 – S4, but as these signals are uncalibrated and may partly or even entirely be caused by interferences from different organic compounds, they are not considered in the results and discussion below.

Line 168: If the signal from HPMTF is significant then the isotopic pattern of the sulfur from HPMTF should dominate over the potential pattern of N2O5. This could provide an experimental constraint on the identity of the 235 m/z species.

In Section 2.3, we add a detailed discussion of this effort, which, however, did not help on with the exclusion of any significant contribution of $N_2O_5$:

> We also tried to explore the isotopologue signatures for the HPMTF and the $N_2O_5$ clusters. The exercise was done for measurement day II. The 237 and 235 m/z peaks (the HPMTF $^{34}$S and $^{32}$S clusters, respectively) show a similar temporal evolution. However, the average ratio found for the areas was very noisy and about 4-fold the expected 4.2%, indicating that the 237 m/z peak is blended by some isobaric unknown ion. Therefore, this method could not be used to ensure the absence or low significance of $N_2O_5$. For the 236 m/z peak (the blended $^{33}$S-HPMTF (0.8%) and $^{15}$NNO$_5$ (0.4%) cluster) no significant peak was detected at all, which due to the generally low signal to noise ratios also cannot rule out some contribution by $N_2O_5$. This applies to days III and IV as well since S/N is very similar.

Line 175: I*SO2 is highly water dependent with the cluster only able to be used if the conditions are dry and the ratio of I*H2O to I are very low. The high and changing RH in the system would make this measurement hard.

We decided to move this mention of the obvious I-CIMS technique, which we did not use, to the end of this paragraph. We also decided not to discuss the water vapor dependence because it is of no relevance to the work presented here:

> The iodide cluster I*SO$_2^-$ could not be detected with our setup, most probably due to the electric fields generated within the ion funnel leading to collision induced dissociation.

Line 177: CO3- can arise from O2- chemistry and ozone. The variability on the SO2 measurement using this technique could be also due to the varying O3 concentration. See Novak et al (https://doi.org/10.5194/amt-13-1887-2020) for an in dept description of the CO3- chemistry and its correlation to O3. This interplay could change the sensitivity of SO2 in the CIMS.

We thank the reviewer for pointing out the ion chemistry discussed in the Novak et al. Paper and potential interferences. However, we do not think that this can be the source of the $CO_3^-$ ions observed in our study. This has been updated to the manuscript:

The possible origin of the $CO_3^-$ ions from $O_2^-$ chemistry in a two-step process involving $CO_2$ and $O_3$ as described by Novak et al. (2020, see R4a/b and R5a/b) also seems rather improbable due to the very low $O_2^-$ signals involved. This is supported by the absence of any correlation of the $CO_3^-$ and $SO_5^-$ signals even for significant changes in $O_3$ as, e.g., encountered for the second ozone addition (see Fig. 2 around 8.5 h). Unfortunately, a detailed pathway for the $CO_3^-$ formation cannot be given.

Line 181: I understand the difficulty in calibrating the sulfur intermediates and understand the use of ncts for the observation of HPMTF and TPA. Were the ncts for HPMTF, TPA, and SO2 background subtracted? Showing the signal as a delta would help illustrate the change in signal over the course of the chamber experiment.

We agree that this point should be further detailed and have added a paragraph separately for $SO_2$ and HPMTF/TPA. This has also induced some restructuring of the CIMS section in order to be easier to follow by the reader:

We referenced all $SO_2$ data to the starting signal (ncts) once humidification and $CO_2$ injection were completed and signals stabilized (days III and IV). For day II due to a late measurement start of the CIMS (just before turning on UV lamps and about one hour after the DMS addition) the reference signal was derived from the very first measurement points somewhat later than for days III and IV. However, the $SO_2$ reference signals which corresponded to about 200 ppt agreed within about +/- 100 ppt for all days. This is well below the other main error sources. We cannot assign the origin of this background. Presented $SO_2$ therefore represented the increase from the start of the DMS oxidation which corresponds to the reacted DMS. Uncertainties resulting from this procedure mainly due to changing humidity and ozone are estimated to a maximum of +50 and -350 ppt of $SO_2$, respectively. However, the comparison to the DOAS instrument shows a quite good agreement of better than 10 % for experiment IV when both measurements were operating (Fig. 4) giving further confidence into the applied procedures.

Calibrations for HPMTF or TPA were not carried out for this campaign, therefore just the ratios of counts for detected product ions over the sum of the $I^-$ and $I*H_2O^-$ signals are reported here. The change in this quantity, here termed normalized counts or ncts, is to first order proportional to concentrations of the neutral analytes (e.g. Huey, 2007). However, missing a proper way to scrub the relevant species detected by the CIMS while retaining ozone and humidity, we could not establish a proper zero background here. For the HPMTF and TPA measurements this procedure did not produce consistent starting values and we report none-background corrected values. These background values before expected presence of OH (UV light for days II and III and $O_3$ for day IV) where below 20% of the maximum observed signals also for day II with the late measurement start of the CIMS. Clearly, we cannot exclude preexisting HPMTF or TPA at these starting points. Therefore, in our view it is also not possible to establish a HPMTF formation from DMS and ozone alone based on the available data. Also, some background variations expected due to variations in relative

humidity, which usually dropped slowly but significantly over the experiments (up to 35% over a 8h measurement, see Supplementary Fig. 2) cannot be ruled out.

Line 183: How much did the humidity drop? Could this also affect the potential for humidity wall driven reactions throughout the chamber experiment? I would reference supp. Figure 1 as it presents the change in RH over the experiment

We have quantified the drops in relative humidity observed over the experiments referring to Supplementary Fig. 2 showing the highest changes (see last lines of the previous reply). We cannot completely rule out that such gradual humidity changes may change wall uptake or desorption but do not think that this will have major effects on the conclusions taken in this work.

Line 204: Looking at Figure S1 it seems that the temperature within the chamber changes by almost 10 C over the time of the experiment and that temperature never seems to reach 25 C. I would add a comment on this and how this much change in temp could alter the observed DMS + OH and the fraction of DMS shutting down either the OH addition and abstraction channels.

In the revised manuscript, we add the following paragraph with a discussion of this temperature dependence to Section 3.1:

Effects from the observed temperature variations of ~10 °C in Experiment I (see Supplementary Figure S1c) on the DMS + OH reactions are impossible to discern due to the low precision of the observed DMS decay and lacking observations of the OH concentration and variability. Nevertheless, it should be noted that while the rate constant of the DMS + OH abstraction reaction has a moderate temperature dependence ($< 5$ % over temperature range of $16 - 35$ °C over all four experiments, see Supplementary Figures S1 – S4), the rate constant of the DMS + OH addition reaction changes from $3.3 \times 10^{-12}$ at 16 °C to $1.3 \times 10^{-12}$ at 35 °C (Burkholder et al., 2019). This means that at the lower end of the temperature range in our experiments, we expect about 60 % to react through abstraction and 40 % through addition, while at the upper end, it is more like 80 % going through abstraction and 20 % through addition. As a consequence, we would expect more OCS production at higher temperatures as we get more HPMTF from getting more of the primary $RO_2$ and a faster isomerization rate due to the temperature.

Line 205: How was dilution determined within the chamber? Were experiments performed elsewhere or was there a dilution tracer added and monitored within the experiment?

The dilution is determined from the volume flow rate into the chamber, and the observed decrease of $CO_2$ that was added to the chamber early in the experiment confirms this. We make a note of this in Section 2.1 in the revised manuscript and show an additional figure to illustrate this in the supplement.

Line 213: Were any experiments done without sulfur (DMS) present while the system was oxidized utilizing the entire array of instrumentation. I am wondering if there is potential for the contribution of contaminants that could lead to prompt formation of OCS. You state that other

VOC (line 208) may have off gassed leading to higher CO, could this lead to a prompt/background OCS?

Prior to the 2020 DMS oxidation experiments described in our paper, the SAPHIR chamber had not been used for any dedicated experiments involving sulfur species. Therefore, an accumulation of significant amounts of OCS or precursors thereof on the chamber walls that would explain the formation of the observed mole fractions can be ruled out.

Line 215: I think it is important to say that the OCS yield is an experimental yield that takes into the environmental parameters. This removes the ease to quickly see this value as a way to simplify the global yield of OCS from DMS oxidation.

Indeed, we do not want to see the yields observed in our experiments to be directly applied in global models or simple budget calculations. We add the following statement at the end of Section 3.1 to make this clear already at this point:

> This, as well the OCS yields given in the following Sections, are experimental results that strongly depend on the conditions in the SAPHIR chamber. We expect OCS yields from DMS oxidation in the real atmosphere to be significantly lower, as will be discussed in Section 4.

Could you also describe how the OCS yield was calculated throughout the chamber experiment? Was the OCS yield calculated from comparing the DMS loss rate to OCS production rate as is shown in the figures or was the yield calculated from the change in signal of DMS and OCS (d[OCS]/d[DMS])? Adding a equation would help illustrate this cleanly.

We now include a detailed description on the yield calculation and an equation:

> This was calculated by dividing the amount of OCS present at the end of the experiment by the dilution corrected total sulfur in the chamber (any sulfur was present as DMS at the beginning of the Experiment, see Supplementary Figure S5) minus the DMS still present:

$$\Phi_{OCS}(t) = \frac{[OCS](t)}{\sum_{DMS\ additions}[DMS]_{init} \times (1 - k_{SAPHIR})^{t-t_{init}} - [DMS](t)} \tag{1}$$

> where $k_{SAPHIR}$ is the observed chamber dilution rate and the summation is carried out for each DMS injection.

Line 236: Dyke et al and others have proposed the reaction of DMS with Cl2 can lead to the formation of CH3SCH2Cl (DMS-Cl). Was there any evidence in your VOC measurements (PTR) that this intermediate arose upon the addition of Cl2? And could this intermediate and its subsequent photolysis or OH oxidation change the OCS yield? Lastly, Urbanski and Wine and Arsene et al found that the DMS + Cl could form an adduct with eventual addition of a O2. Is there any evidence in your PTR measurements of additional OH-addition like chemistry? (i.e. increased DSMO/MSIA/DMSO2) Shallcross et al 2006 AtmEnv; Copeland et al 2013 EST; https://doi.org/10.1021/jp992682m (Wine and Urbanski)

In the PTR-MS, $CH_3SCH_2Cl$ would be expected to show up at m/z 97 and 99 for the two main Cl isotopes. At neither of these m/z ratios, a strong signal or a pattern suggesting a mechanistic relation to our experiments are observed.

As stated above (in response to the comment re line 140), there are traces in the PTR-MS data that could be related to DMSO and $DMSO_2$, and they are now shown in the supplementary Figures. But as even the qualitative assignment of these m/z ratios bear substantial uncertainty, we feel that the derivation of any mechanistic details from these data gets deep into the realm of speculation.

Similarly, the relative contribution of the reactions DMS + OH, DMS + Cl and DMS + $Cl_2$ to the DMS loss in Experiment III can only be speculated upon, because there is no quantitative information on the concentration of Cl radicals. Dark experiments with DMS + $Cl_2$ are needed to confirm the published rate coefficient and products, but were unfortunately not carried out in our 2020 SAPHIR experiments.

Line 243: The temperature in the chamber is greater than 298K (S3 reads about ~305K). This elevated temperature would effect the fraction DMS oxidizing by the OH additional and abstraction as well as the isomerization rate. A comment about the variability in temperature across all experiments and if the variability in temperature would affect the product distribution would be greatly appreciated. OH-addition has a strong temperature dependence and as such would greatly change the amount of DMS shuttling down HPMTF and OCS.

Such a discussion is now included in Section 3.1 as already described in response to the comment in line 204.

Line 252: Barnes et al 1991 (https://doi.org/10.1002/kin.550230704) found that ClO can react with DMS at a rate constant of $(9.5 \pm 2.0) \times 10^{-15}\, cm^3\, molecule^{-1}\, s^{-1}$. I think it is important to cite this finding as support for the idea that this reaction could compete.

The citation will be added here.

They also suggest that the reaction would lead to the formation of DMSO, shutting of the potential production of HPMTF and OCS. If possible a small discussion/observations of chloride chemistry would be appreciated. Iodide CIMS has the potential to observe multiple halogens (i.e. Cl2, ClO, ClNO2). Do you have any observations of other halogen species (ClO, ClNO2) that could help understand the lower observed OCS yield and the mechanisms driving the DMS oxidation? With Cl2, JCl and O3 observations could you model/approximate the [ClO]?

With the setup employed we unfortunately cannot detect HCl, $ClNO_2$ and ClO since the appropriate cluster ions fall apart in our ion funnel and low iodide CIMS sensitivity in case of ClO.

We add two sentences describing what role of ClO in the initial DMS oxidation step we would expect:

It should be noted that the Cl + $O_3$ reaction is about ten times slower than the Cl + DMS reaction and a maximum ClO concentration of 4 x $10^7$ molecules $cm^{-3}$ is estimated from

steady state calculations. Because ClO is less reactive towards DMS than Cl (ClO reacts with DMS at a rate constant of $9.5 \pm 2.0 \times 10^{-15}$ cm$^3$ molecule$^{-1}$ s$^{-1}$ according to Barnes et al., 1991), it is not expected to be competitive with Cl and even Cl$_2$ for the oxidation of DMS.

Line 275: Could the drop in HPMTF also be associated with increased NO and thus less HPMTF (e.g RO2 + NO dominates/increases over isomerization). S3 shows a significant step change in NOx with a subsequent sustained concentration of O3 leading one to question the start of NOx cycling. Could NO chemistry start upon the opening of the roof?

Similar levels of NO and NO$_2$ were also present in Experiment III and much of the DMS has already been removed by the time the roof is opened. Therefore, we would not expect HPMTF production to dominate the HPMTF trace here, and the drop of HPMTF is more consistent with a faster/additional removal process such as photolysis triggered by opening the roof. We add the following sentence in the revised version:

As much of the DMS has already been removed by this time, this drop would be difficult to explain by a sudden change in the HPMTF production rate (e.g., from competing binary reactions) and is more likely caused by a faster/additional removal process such as daylight photolysis. This is, however, only speculative, …

Line 276: Could an approximate photolysis term for HPMTF be calculated using the decay in HPMTF? Using the assumption that HPMTF + OH is 1-2E-11 from Jernigan et al and Ye et al, could an approximate photolysis rate be determined from the missing HPMTF loss? This may be an oversimplification of the observations, but an approximate vale could be compared to the values used in Khan et al to determine the weight of HPMTF photolysis and its potential roll in OCS formation.

We do not think that this is possible. HPMTF photolysis would only set in after the roof is opened, and the shape of the observed trace suggests a faster HPMTF decay compared to the earlier experiments already before the roof is opened.

Line 289: The clarifications done here are greatly appreciated in showing the complexity of OCS from DMS oxidation and the use of mechanism over a yield. I would still recommend the use of "experimental" or "chamber" as a clarifier for the observed OCS yields.

We appreciate this recommendation and will do so.

All Figures: If the OCS yield is calculated by relating the rates of DMS loss and OCS production, then I would recommend adding the yield as a function of time in the experiment. This could help show if the value is constant or variable depending on the various perturbations. I can also see that DMS and OCS may not be tightly correlated given the need to transition though intermediates, so this time dependent yield could be misleading. Any comments on this would be appreciated.

Rates are approximated by fitting slopes of the DMS and OCS traces over different time windows, and the time resolution of the OCS rates is too low to adequately reflect changes or events at certain points in the experiments. Therefore, as explained above, the yields were

determined only for total OCS produced by the end of the experiment, and this information is now included in Section 3.1.

Figure 1: Do any species respond to the second addition of ozone? Upon first glance nothing in the SI or figure responds to the increased O3 and subsequent OH formation?

A small increase in $RO_2$ and $HO_2$ is observed (Figure S1a), and the first order DMS removal rate increases from about 8 to about 12 (Figure 1d), although with a large uncertainty due to the gap in DMS data just prior to the $O_3$ addition. We would not expect more dramatic or sharper changes because i) there is likely to be an excess in oxidant already and ii) the chamber response will take time because OH is only produced in the vicinity of the UV lamps.

Figure 3: There seems to be an abrupt and unexplained drop in Cl2 at hour ~4.6 after DMS injection. Could you please explain what occurred at this hour and if that abrupt drop and loss of chloride could be used to understand a transition between Cl and OH dominated oxidation?

The drop in $Cl_2$ is entirely expected as it coincides with the stop of the $Cl_2$ addition. This information is given in Table 1 and in line 125, but we will repeat it in Section 3.3 and add a bar showing the addition period in the figure.

Technical comments

Line 165: Define TPA

We add the full name in the revised manuscript.

Figures: I have trouble reading the axis titles, especially the multiplier (i.e. $10^{-5}$). Could you please increase the font or resolution of the figures.

We will rework all figures to improve quality and readability.

Figure 1: Which measurement for HCOH was used here. The table states two instruments were used. Please add for this and all future figures if the concentrations are dilution corrected.

The HCHO measurement shown is from the CRD instrument, and we will add this information to all relevant figure legends.

All measured concentrations are given 'as observed', i.e. without applying any correction. For observed species, any dilution effects are included by definition. Only for calculated parameters such as OCS yields and the total amount of sulfur shown in Figure S6, the dilution effect is actually calculated.

Figure 4: I recommend adding labels for the three modeled lines in 4c. I see that the description is in the description, but a label will help guide the eye and discern colors.

Such labels are added in the revised version.

---

## Author Comment (AC2)

We thank the reviewer for the positive feedback and the constructive comments that we address point by point below. The review comments are repeated here in black, our replies are given in blue.

Major comments:

1. The figures all have low resolutions. It is difficult to read the superscripts in the axis legend. Figures of a better quality need to be provided.

The quality of all figures is improved in the revised manuscript.

2. Are there any losses of precursor or products to the chamber wall?

We do not expect significant loss of the precursors and products shown on the FEP walls of the chamber at the selected/observed concentrations on the time scales of the experiment, and we do not see any evidence for such deposition processes in our results.

3. In terms of the sulfur budget, were there any aerosol formations in these experiments?

Formation of sulfate aerosol during our experiments can't be ruled out. However, it is neither supported by the observed sulfur budget (Figure S6) nor expected under the experimental conditions. Note that during previous SAPHIR experiments, in the low concentration range used in our experiments, particle formation was never observed without the introduction of aerosol seeds (Carlsson et al., 2023; Brownwood et al., 2021).

4. In Figure 2, it seems that DMS started to decay before the UV light was turned on. Is the decay from DMS+$O_3$? The figure also suggests that HPMTF formation started before light was on. Does this indicate HPMTF can also be generated from DMS+$O_3$?

The observation that DMS starts to decay before turning on the UV lamps is, indeed, one of the mysteries in our experiment, and we have spent some time discussing this. The chemical model does expect DMS reactivity to be dominated by $O_3$ here, but the rate is too low to explain the observed decay. Even with an enhanced DMS + $O_3$ rate constant of ~1 x $10^{-18}$ observed at high relative humidities by Wang (2013), the expected DMS loss rate is about a factor of 10 slower than observed. Only if DMS + $O_3$ in the aqueous phase (kinetics were reported by Hoffmann et al., 2016) is considered, the observed DMS loss can be explained, e. g. in a water film forming on the cold chamber walls in the morning upon humidification. But this is really speculation and we would rather not include this in the paper.

With respect to HPMTF generation from DMS + $O_3$, we would not interpret the non-zero HPMTF levels as evidence for HPMTF presence or formation before the light is turned on, although we can also not exclude this. The difficulties of establishing a proper background for HPMTF and TPA are explained in some detail in the response to a comment by Reviewer #1 (re. line 181) and we can only report none-background corrected values. Therefore, it is not possible to establish a HPMTF formation from DMS and ozone from our observations.

Technical comments:

5. Line 546, should be "OCS production rates given in panel d)".

Thanks for spotting this, we correct it in all captions of Figures 2 – 4.

6. The OCS time series are quite fluctuated. Did the authors calculate OCS production rates using a longer period of time and what is the difference?

The fitting of the OCS production rates is explained in the figure captions. Fits were always made over 30-minute time periods as a compromise between time resolution and precision of the calculated rates. The instrument precision and, in case of Experiment I, gaps in the data acquisition, would not allow for meaningful results over shorter time periods. For experiments II – IV, fits over longer time periods yield essentially the same results as averaging the 30-minute rate values, as one would expect.

**References:**

Brownwood, B., Turdziladze, A., Hohaus, T., Wu, R., Mentel, T. F., Carlsson, P. T. M., Tsiligiannis, E., Hallquist, M., Andres, S., Hantschke, L., Reimer, D., Rohrer, F., Tillmann, R., Winter, B., Liebmann, J., Brown, S. S., Kiendler-Scharr, A., Novelli, A., Fuchs, H., and Fry, J. L.: Gas-Particle Partitioning and SOA Yields of Organonitrate Products from NO3-Initiated Oxidation of Isoprene under Varied Chemical Regimes, ACS Earth and Space Chemistry, 5, 785-800, https://doi.org/10.1021/acsearthspacechem.0c00311, 2021.

Carlsson, P. T. M., Vereecken, L., Novelli, A., Bernard, F., Brown, S. S., Brownwood, B., Cho, C., Crowley, J. N., Dewald, P., Edwards, P. M., Friedrich, N., Fry, J. L., Hallquist, M., Hantschke, L., Hohaus, T., Kang, S., Liebmann, J., Mayhew, A. W., Mentel, T., Reimer, D., Rohrer, F., Shenolikar, J., Tillmann, R., Tsiligiannis, E., Wu, R., Wahner, A., Kiendler-Scharr, A., and Fuchs, H.: Comparison of isoprene chemical mechanisms under atmospheric night-time conditions in chamber experiments: evidence of hydroperoxy aldehydes and epoxy products from NO3 oxidation, Atmos. Chem. Phys., 23, 3147–3180, https://doi.org/10.5194/acp-23-3147-2023, 2023.

Hoffmann, E. H., Tilgner, A., Schrödner, R., Bräuer, P., Wolke, R., and Herrmann, H.: An advanced modeling study on the impacts and atmospheric implications of multiphase dimethyl sulfide chemistry, P. Natl. Acad. Sci. USA, 113, 11776–11781, https://doi.org/10.1073/pnas.1606320113, 2016.

Wang, H. Ozone kinetics of dimethyl sulfide in the presence of water vapor. Front. Environ. Sci. Eng. 7, 833–835, https://doi.org/10.1007/s11783-013-0570-8, 2013.